# Professionals’ and Patients’ Perspectives on Criteria for Referring Hypertensive Patients to Comprehensive Medication Management Services in Public Primary Health Care

**DOI:** 10.3390/ijerph20075233

**Published:** 2023-03-23

**Authors:** Maria do Carmo Vilas Boas Sousa, Elizabeth do Nascimento, Simone de Araújo Medina Mendonça, Clarice Chemello

**Affiliations:** 1Faculty of Pharmacy, Federal University of Minas Gerais, Avenida Presidente Antônio Carlos, 6627, Pampulha, Belo Horizonte 31270-901, MG, Brazil; 2Department of Psychology, Faculty of Philosophy and Human Sciences, Federal University of Minas Gerais, Avenida Presidente Antônio Carlos, 6627, Pampulha, Belo Horizonte 31270-901, MG, Brazil

**Keywords:** comprehensive medication management services, patient selection, primary health care, qualitative research

## Abstract

Patient prioritization in comprehensive medication management services allows coordinating care and guiding patients according to their clinical profile and their medication use. The aim of the study is to identify and describe factors that indicate the need for comprehensive medication management services among primary care hypertension patients within a public health system from the perspective of patients, pharmacists, nurses and physicians. A qualitative study was carried out with interviews with nurses, pharmacists and physicians (n = 20), and two focus groups with hypertensive patients (n = 12) at primary health care facilities and a public outdoor fitness area between January and February 2019 in Brazil. All interviews were transcribed and analyzed using the Atlas.ti^®^ software. The data analysis revealed the following factors indicative of the need to refer hypertension patients to a pharmacist: lifestyle habits, comorbidities, health care utilization and medication use. The issues identified and the information obtained from the qualitative research and compared with literature studies reviewed allowed defining dimensions that should be considered as an aid in the selection of uncontrolled hypertensive patients for comprehensive medication management services.

## 1. Introduction

Within the scope of the public health service in Brazil, pharmaceutical assistance is configured in such a way that the pharmacist needs to juggle technical-managerial and assistance activities. The high demand for managerial functions, the lack of physical structure and human resources in health services and the insufficient management support offered are some of the challenges faced by pharmacists, hence hindering their inclusion and/or participation in direct and effective patient care [1,2,3,4,5,6].

Aside from how the service is structured, the high prevalence of chronic non-communicable diseases (CNCDs), which represent one of the main public health issues today, affects the health care service demands [7,8,9]. Among the CNCDs, systemic arterial hypertension (SAH) stands out, being a serious public health issue in Brazil, whose prevalence varies between 22% and 44% among adults (32% on average), reaching more than 50% among 60–69-year-old individuals and 75% among individuals over 70 years old [10]. In 2017, SAH showed a prevalence of 24.3% among adults over the age of eighteen [11].

In this context, considering the pharmacist’s work and the high prevalence of CNCDs, guidelines that contribute to selecting members of the public who would benefit most from comprehensive medication management (CMM) services and to guiding health team professionals on patient referrals to pharmacists are necessary. The service when carried out based on the risk classification allows for care coordination and patient guidance according to the patient’s clinical profile and medicine use [12].

As seen in a wide-scope review [13], there are few validated tools for outpatient selection. The fact that they were not validated demonstrates that it is necessary to improve the development of tools that can accurately identify patients who would benefit from CMM.

To establish patient selection criteria for comprehensive medication management services, it is crucial to know the individual needs and the health care utilization profile. Health services grounded on the population’s needs tend to be more efficient, with a greater ability to listen to the people and provide care according to their health demands [14,15,16,17].

The population’s needs can be classified into visible and invisible. Visible needs are the organic conditions of the body, which guide the individual’s initial search for health care services. Invisible needs are the subjectivity and singularities of every individual [18]. Therefore, understanding the sociocultural pluralities and epidemiological dynamics that subjectively, individually and collectively affect the experiences and realities of patients and professionals is necessary to direct care and improve its quality [19,20].

Qualitative research plays a key role in the process of understanding the population’s needs, as it embraces the patients’ and health professionals’ diversity of knowledge and experiences when it looks at both the individual and collective context [21,22,23] (pp. 11–21). Awareness of patients’ clinical needs makes it possible to draw up an instrument capable of effectively directing the care provided by the pharmacist. Therefore, such an approach will be vital to support the development of an instrument for the selection of hypertension patients for medication management with the pharmacist. Thus, this study aims to identify and describe factors that indicate the need for comprehensive medication management services among patients with hypertension in primary health care (PHC) from the patients’, pharmacists’, nurses’ and physicians’ perspectives to develop a selection instrument for these services.

## 2. Materials and Methods

The study was carried out in PHC facilities and an outdoor fitness area in the city of Belo Horizonte, Brazil. The primary level of health care comprises a network of health centers, which are the population’s main gateway to the public health system. This system is divided into nine regions and has 152 PHC facilities [24].

A qualitative study conducted through semi-structured interviews with health professionals and focus groups with hypertensive patients cared for in a primary health care facility and an outdoor fitness area (this is specifically a public space dedicated to health promotion and care delivery by qualified professionals and part of the *Programa Academia da Saúde*—literally, Health Gym Program) [23,25] (pp. 11–21).

Physicians, nurses and pharmacists were interviewed between January and February 2019. Professionals were selected by convenience sampling, and the number of interviews was defined during the course of data collection, until theoretical saturation. Those selected were emailed or text-messaged an invitation to participate in the study, and interviews were also scheduled through these means.

Interviews were conducted with the help of a topic guide containing questions about the participants’ experience with caring for hypertensive patients, with sharing care, and about the criteria they apply for patient referral to other professionals in the health team.

Two focus groups were carried out with patients aged 18 years or older with a diagnosis of SAH, who had decompensated clinical conditions and/or who were receiving comprehensive medication management services.

Patients were invited through a letter, delivered by community health agents, which contained the objectives of the study and explained the importance of their participation. Participation in the group was confirmed by a telephone call made prior to scheduling dates.

To characterize the profile of the group, participants completed a questionnaire covering personal, sociodemographic and professional information.

The focus groups were conducted by a moderator with the help of a topic guide, accompanied by a co-moderator and an observer. The topic guide included questions about the patient’s knowledge of hypertension, changes in their daily lives after diagnosis, chronic medication use and the need for comprehensive medication management services.

One of the groups was carried out at a PHC facility and the other at an outdoor fitness area (of the Programa Academia da Saúde) in March and April 2019, respectively.

The audio-recorded information was transcribed and then analyzed and systematized using the Atlas.ti^®^ software. Content analysis [26] (pp. 33–52) was used to identify the most relevant factors used in the selection of uncontrolled hypertensive patients, the characteristics that justify the need for comprehensive medication management services and the reasons for patient referral to this service.

The project was approved by the Research Ethics Committee of the Federal University of Minas Gerais (COEP-UFMG) (approval nº CAAE 27284214.7.0000.5149) and by the Belo Horizonte Municipal Health Department (SMSA-BH) (approval nº CAAE 27284214.7.3001.5140). All participants provided verbal and written consent to participate in the study.

## 3. Results

Thirty-two people participated in the study, comprising 20 health professionals and 12 patients. Among the health professionals, there were seven pharmacists, six nurses and seven physicians. Of these, 70.0% (n = 14) were female, 38.4 ± 9.3 years old, and had been working in primary care for 9.0 ± 6.4 years. The duration of the interviews was 25 ± 10 min. Among the patients, all of them had hypertension, 58.3% (n = 7) were women and 58.3% (n = 7) were married. In terms of schooling, most of the patients, 58.3% (n = 7), had not completed elementary education and only 16.7% (n = 2) had completed high school. The average duration of the focus groups was 45 min.

Several topics emerged from the data analysis concerning the characteristics underlying patient referral to a pharmacist for assistance. Based on the data obtained from health professionals and hypertensive patients, common factors were identified and grouped into four dimensions, namely lifestyle habits, comorbidities, health care utilization and medicine use.

The Table 1 summarizes the main characteristics identified in this study.

According to the metrics obtained in the data analysis with Atlas.ti^®^, the number of excerpts coded by theme is shown in Table 2. Other themes were identified; however, they were not relevant in relation to the research aims; therefore, they were not included in the study results.

### 3.1. Lifestyle Habits

In the first dimension, different topics were considered, such as smoking, alcohol consumption, physical activity and salt consumption. The objective was to evaluate and identify the main factors capable of influencing blood pressure (BP) control in patients with hypertension.

The interviewed health professionals addressed some major characteristics related to lifestyle habits that can influence BP control. Among them are smoking, sedentary lifestyle and inadequate diet, with emphasis on excessive sodium consumption and alcohol intake.

The interviewed patients also emphasized some of those aspects, such as reducing their daily salt intake. During the focus groups, they talked about the doubts on the concomitant use of alcoholic beverages and medications. This kind of uncertainty, according to the interviewed health professionals, is frequent, hence demonstrating the need for educational campaigns to raise awareness among the population. These topics are highlighted below.


*(...) they tell us to reduce the amount of salt in our food. So, we have to cut down on sodium, as sodium is the worst part of it, which is found in salt. So, we have to cut down on everything.*
(patient 1)


*It’s funny... I was told something else, like if you’re going to drink, don’t forget to take your medicine... not in my case, in my husband’s case, as he takes it too... so, well, he takes the medicine and drinks... but not every day.*
(patient 7)


*We’ve seen examples of people who (...) like to drink beer, and occasionally, on weekends, for example, drink beer and don’t take their medicine. So, such myths still exist. It turns out that, because of such a behavior, they skip doing their treatment, which is supposed to be a continuous treatment.*
(pharmacist 4)

As for the relationship between lifestyle habits and illness, the professionals addressed changes that arise from the treatment of the disease, and prevention of future events. From their viewpoint, the consequences and worsening of patients’ health status could be minimized if preventive measures were implemented before the disease appears or at its outset. The pharmacist emphasizes the importance of patient education about their lifestyle habits, health problems and medication use.


*So, a healthy lifestyle is very much associated with illness in the population that we care for today in general. So, when you say you need to cut down on sugar, reduce salt intake, avoid canned foods, seek regular physical activity, you’re already associating this with a disease picture and not a prevention picture. So, it’s much harder to encourage healthy living habits in this population because of that… because it’s hard for you to make them change habits.*
(pharmacist 3)

### 3.2. Comorbidities

In this dimension, information on the patients’ and their families’ past and current clinical history that can affect BP control was collected. Some diseases that may be directly or indirectly related to BP control were also included.

Among the comorbidities brought about by the health professionals, the prominent ones were obesity, diabetes mellitus, mental health, dyslipidemia, congestive heart failure, angina, coronary artery disease, nephropathy, kidney disease, rheumatologic disorders, endocrine disorders and liver disease. Furthermore, some of these clinical conditions can lead to complications, such as atherosclerosis, coronary heart disease, cerebrovascular accident (CVA), heart attack and others.

During the focus group session, participants talked about their experiences with close relatives who experienced complications resulting from uncontrolled BP, such as stroke and kidney problems. In the excerpt below, a participant related a personal experience about high blood pressure in their parents. *(...) my mother’s* [blood pressure] *has reached 270 over 180. She had two strokes but no sequelae. And last time, it was 270 over 180 again... and she was inside the hospital. So, we do have a history; mine is not high, but the whole family* [has] *high blood pressure. Her kidneys ‘shut down’.* (patient 5)

As was observed in the study, not following the pharmacological treatment can cause damage to target organs and influence the patient’s quality of life, leading to an even more complex medication management. This was highlighted by a physician who treats patients with hypertension.

*Changes in renal function. So, when the kidney starts to show signs of failure, it is very hard to control it. You start having to give* [the patient] *high doses* [of medication], *involving several combinations* [of medications]. *So, kidneys are a real warning sign.*(physician 1)

One of the focus group participants reported complications due to non-treated and uncontrolled high blood pressure, which led him to undergo hemodialysis for a long time and later need a transplant of both kidneys.


*I had hypertension, (...) but I stopped taking the medication. I was younger. It started worsening and worsening..., it got way worse. I ended up at the doctor’s, and I’m still undergoing treatment... I lost both kidneys, I received a transplant, I underwent hemodialysis for a long time because of blood pressure issues... (...) I was diagnosed with high blood pressure, that’s what’s in the medical report. I remain in treatment until today.*
(patient 4)

### 3.3. Health Care Utilization

Knowing how patients are inserted in health services and how they use and interact with the care provided allows outlining actions that meet their real needs. This study identified characteristics that encompass and allow assessing some of the factors affecting the access and use of such services by patients, from different perspectives. Another important aspect discussed is their search for outpatient or emergency medical care due to uncontrolled BP issues. One of the pharmacists interviewed pointed out that the patient does not have access to some medications in certain situations, as noted below.

*They* [patients] *will not have access to prescribed medicines for a number of reasons. Sometimes, they will not have access to other government programs, such as the Popular Pharmacy Program, for just not knowing about it, and they just can’t bear the costs of their treatment. It’s not always like that, but we often lack anti-hypertensive medication. I think that it’s a factor that undermines the treatment. (...) Sometimes it’s related to use. As I said, the lack of medication is an issue, or the prescription of non-standardized medication... sometimes, it’s because the medicine is expensive, so the individual can’t afford to buy it, or due to adverse reactions…*(pharmacist 4)

Different factors can affect access to medicines, such as their unavailability in PHC facilities or lack of access to essential medicines, for instance. In addition to the inappropriate use of medication in different aspects, whether individual, social or economic, these factors may influence the search for care at a health care facility. The interviewed professionals reported that some patients used the health services more frequently because of uncontrolled BP.

[a patient] *who is out of therapeutic goals, and this can mean a patient who attends the health facility too often, who is repeatedly medicated in the observation room…. or a patient who (...) reports that s/he doesn’t use the medication, (...) who gets blood pressure checked and then leaves. (...) Just like I told you, these patients who are repeatedly polymedicated in the observation room are here every day and we know why the pharmacist always reports: “Every day I’m dispensing captopril to Ms. So-and-so”.*(pharmacist 3)

Such statements can be confirmed by a patient who recounts his experience with uncontrolled hypertension and his need to resort to health services frequently to obtain his BP checked at the health facility. Moreover, he narrates that he once had to use an emergency service due to uncontrolled BP. It is worth noting that, in his case, some situations were not considered in depth to prevent this type of fact from being recurrent.

*Mine* [blood pressure] *(...) is a little better now, but some days ago, I’ve been feeling pretty bad with high blood pressure. My blood pressure had gone up to 190, 200. (...) they told me to go to the emergency care unit. I’m always, always here getting my blood pressure checked, almost every week. I even have the blood pressure log that the doctors tell me to bring along. And every time I come here to fill it in, my blood pressure is still high.*(patient 3)

### 3.4. Medication Use

The last dimension that arose from the interviews and focus groups comprises factors related to the use of medicines that can affect blood pressure control. It covers objective questions, such as how to use the prescribed medication, the quantity used and how many times a day it should be taken, in addition to particularities of the patients’ routine, such as changes in the medication administration timing to suit their daily life and work routine. Polypharmacy, medication regimen complexity, difficulty in understanding the medical prescription and self-medication were related to incorrect adherence to and use of medicines, in addition to the occurrence of adverse reactions, which can cause the patient to stop taking the medicine. Changes in the patients’ daily routine or the need to perform activities outside the home cause them to not take their medicines properly, especially diuretics.

One recurrent situation is the addition of new medicines to the patients’ prescription when the physician notes that their blood pressure levels are not adequate; however, it is paramount to check how the patients are using the medicines before including others. It is known that there is a verticalization in the physician–patient relationship, and often times the patient may feel embarrassed and thus not report their difficulties to the physician. Pharmacist 5 describes his experience concerning cases in which the patient had difficulty controlling hypertension. In some cases, other medications were prescribed without investigating the possible causes of uncontrolled blood pressure, as observed below.


*(...) sometimes, there’s all that pharmacotherapy to follow. The patient just can’t take all the medicines properly. But the doctor doesn’t understand that… and just prescribes some more to that pharmacotherapy. (...) It turns into a snowball, it gets complicated, and the patient will take them all wrong. Sometimes s/he has hypotension, a fall, and all that stuff. So, I think this all interferes.*
(pharmacist 5)

One factor that can affect the correct use of medication is the patients’ understanding of their own treatment. In many situations, they do not understand how they should use the prescribed medicines. Therefore, it is essential to reinforce the information on dosage, and ask patients to repeat the explanation received on how the medicines should be used. This check helps to identify possible misunderstandings so that the treatment is effective and safe. One of the physicians described the difficulties that patients have to understand concerning the medication use.


*Another thing we see is when the patient has cognitive issues. When you realize that s/he is not understanding the treatment or when you realize... that it’s the classic patient who, when you ask them “what medicines are you taking?” and s/he just doesn’t know how to answer you. When you ask them which ones... if s/he can tell you about the medicines s/he takes and at what time… it’s the one who doesn’t know how to answer you; s/he’s not taking the medicine properly.*
(physician 2)

Medication regimen complexity when associated with the patient’s routine activities impairs and/or hinders adherence to treatment. Therefore, the patient’s routines need to be considered in prescription, thus adapting the medication timing to their daily lives. Some focus group participants reported their difficulties to remember their medicine dosage, a fact that corroborates what the interviewed professionals stated. The patients’ statements demonstrated the importance of considering their experiences with medicine use during comprehensive medication management services in order to understand each person’s subjectivity and needs, and therefore adapt the pharmacological treatment to their reality.


*Oh, I don’t take it on time either. I make my own schedule. All of us forget it. Me, when I remember to take it, the right time is gone, then it’s only the next day.*
(patient 5)


*The only medicine I forget to take is the one for diabetes..., which I have to take in the morning, at lunch and dinner time... it’s the only one I forget.... but the other ones it’s just in the morning, so I don’t forget about it...*
(patient 4)

Another aspect highlighted by the health professionals was self-medication, which, when not guided, can bring health risks. One of the patients reported that she had adverse reactions while taking medication on her own. This points out the importance of providing patients with orientation on the positive and negative factors of self-medication, with information that will be useful when they self-medicate.


*I almost developed Alzheimer’s by mixing two meds on my own. I almost went 15 days without sleep.*
(patient 5)

The emergence of adverse reactions must be monitored to be avoided, and if they occur, they must be identified early so that the treatment effectiveness is not impaired, as described by the patient 3.

*(...) a medication swelled me up completely, I used to take it. And my feet got swollen, and felt like balloons... my legs swelled, my face swelled up. Then they* [doctors] *told me to discontinue it* [amlodipine], *I stopped taking it and no more swelling.*(patient 3)

Patients brought about some external factors that can also influence the BP levels, causing them to go up, such as the appointments with health professionals and concerns with everyday matters. In addition, the anxiety generated prior to BP checking also affects its levels, and this causes patients to fail to attend the checking appointment, which may represent a risk to their health, as can be seen in the following excerpt

*The higher it* [blood pressure] *gets,* [the more] *I get worried, and it goes up even more. So I myself don’t check it, I just don’t. Only when I come here do they get it checked, and it’s high. Sometimes they refer me to floor below* [the observation room]. *Sometimes they refer me to the hospital…*(patient 1)

Despite being a factor that is beyond the patient’s control, personal, family and social matters indirectly interfere with their health, mainly regarding their blood pressure control. In this regard, thinking about the patient’s environmental context can help to propose measures that contribute to the adequate and effective control of hypertension in a personalized way, as was emphasized by some health professionals interviewed.

*Besides, it’s also about the community s/he* [the patient] *is inserted in. How does s/he… how does this community interfere…. What if s/he is elderly, if s/he takes care of the family, if s/he is the breadwinner, how much time does s/he have to take care of his/her own health. (...) What are the other demands of his/her life, because sometimes these are much more important for his/her evaluation than repeated appointments, for an adequate BP control. So, it all does influence. Sometimes s/he’s in an area of risk, the level of stress. Large families, with little income… maybe this creates great stress. And stress will interfere with blood pressure control. Sometimes, we just look at the anti-hypertensive medicines and fail to look at such a family context. So it’s all interrelated.*(pharmacist 3)


*Can the person read, or can’t s/he? S/he will need you to… you will have to create some identification, because s/he cannot read, but s/he has to take it [the medicine]. If I need to, I’ll draw it. You need to come up with something. Does s/he have a caregiver to help... is s/he elderly?*
(nurse 1)

In addition to the social aspect, it was observed in this study that the lack of family support can influence SAH control, which may negatively affect the treatment thereof.

The pharmacist may also identify patients who have a problem with the pharmacological treatment, as some of them, during dispensing, refuse to receive medication at the health facility’s pharmacy. Once this situation is identified, the pharmacist is able to propose actions that will bring about improvements regarding medicine use.

*(...) a patient in line at the pharmacy* [for example], *when it’s his/her turn to get the medicines, s/he refuses some of them. I think this can also be a warning sign for us to carry out a medication management assessment of this individual’s experience with his/her medicines and investigate the control or non-control of blood pressure, as there in the line, at that moment, we won’t know whether it’s controlled or not. But I think it’s a factor that really sets off some alarms.*(pharmacist 4)

Given the above, it was observed in this study that the participating patients and health professionals brought about similar and complementary aspects and perceptions of factors that influence patient referral to comprehensive medication management services. The knowledge generated may help draw up an instrument for the selection of patients who really need such a clinical service.

## 4. Discussion

This study allowed identifying, based on the participants’ experiences, factors that can help in referring hypertensive patients to comprehensive medication management services. The topics covered by the participants are in line with topics addressed by other studies that developed tools for the selection of patients to be cared for by a pharmacist [13,27,28,29,30].

Lifestyle habits strongly influence the development and control of hypertension. Inadequate sodium consumption, improper diet and insufficient physical activity are some of the factors that can lead to hypertension and maintain uncontrolled BP levels. In this way, health professionals can help raise their patients’ awareness of the necessary care for the treatment of hypertension and the risks and complications that result from inadequate BP levels. Medical literature shows that lifestyle changes, such as regular physical activity, weight loss and healthy eating, combined with the use of medications, can enhance the effectiveness of the treatment, and consequently reduce high BP levels. In addition, smoking and alcohol cessation contributes to reducing cardiovascular risk. Health professionals play a key role in promoting adherence to behavioral changes that can help control hypertension [30,31,32,33,34]. Therefore, the identification of lifestyle habits that interfere with BP control is crucial for the selection of patients for referral to comprehensive medication management services.

Comorbidities added to inadequate lifestyle habits contribute to uncontrolled BP, and therefore need to be considered when selecting patients for clinical care by the pharmacist. Patients with uncontrolled BP levels are at increased risk of damage to target organs, such as kidneys, heart and brain. Studies show that poorly controlled hypertension is associated with mortality, increased cardiovascular risk and disease progression. Factors such as advanced age and obesity also contribute to the occurrence of clinical complications in hypertensive patients [34,35,36,37]. Therefore, these aspects need to be considered when proposing a patient care plan, as well as other factors such as the patients’ values and culture, subjective experiences with their clinical conditions and medicine use, for the achievement of therapeutic goals [30].

The frequency of patient use of health services may be associated with lack of clinical parameters, exacerbation of chronic diseases and comorbidities. Therefore, knowing the demands and needs of the population assisted in a health care facility is paramount to proposing measures and actions so that care is more effective and conclusive for the population. It is worth emphasizing that many diseases are considered ambulatory care sensitive conditions (ACSCs) and therefore PHC plays a vital role in the management and control of such conditions, which, when treated early and effectively, prevent complications and hospital admissions [15,38,39,40].

Finally, the medication use also impacts the control of blood pressure levels. Thus, identifying whether there are problems arising from the use of medicines is essential when selecting patients for referral to a pharmacist. In most cases, patients with hypertension have multiple morbidities and therefore need to use several medicines. This increases the medication regimen complexity, which is associated with the characteristics of the prescribed treatment and the patient’s clinical conditions, and consequently the expected therapeutic goals can be negatively affected. Furthermore, the dearth of mechanisms that allow disseminating and exchanging experiences and information between health professionals and patients may lead to low effectiveness of health-oriented actions [41].

It is noteworthy that the four dimensions addressed are codependent and influence one another. Thinking of the patients in a holistic way, i.e., considering their personal beliefs, values and culture, can help achieve the therapeutic goals [17,18,30].

The participation of patients and health professionals in the process of obtaining information about SAH will favor the devising of an instrument capable of identifying, in a comprehensive and multidisciplinary way, hypertensive patients who need medication management.

### Strengths and Limitations

This study relied on health professionals (pharmacists, nurses and physicians) who work in primary care facilities, know the profile of patients seen and are aware of the main aspects related to SAH and medicine use. The participation of hypertensive patients contributed to broadening the view about such aspects based on their perceptions. In this way, a relationship between the patient’s and the professional’s views can be established, and it is important to develop an instrument that includes these perspectives.

As limitations we cite: the study was carried out in just one city and with a small number of patients and health professionals and the instrument is aimed at solely one service (comprehensive medication management services) in primary care.

## 5. Conclusions

Qualitative data obtained through interviews and focus groups and compared to the current scientific literature enabled the identification of major factors influencing the demand for care provided by pharmacists from the perspective and perception of health professionals and patients regarding SAH and its treatment. The information brought about via this study will be essential to creating an instrument for health professionals seeing hypertensive patients in PHC, aimed at directing the care delivered by pharmacists. More studies are necessary to confirm and validate the data obtained in this research.

## Figures and Tables

**Table 1 ijerph-20-05233-t001:** Summary of the main themes identified in this study.

Themes and Sub-Themes
*Lifestyle habits* Practice of physical activity. Consumption of salt. Cigarette use. Alcohol use.
*Comorbidity* Presence of other diseases.
*Health care utilization* Access to medical appointments. Access to health services. Access to medication. Hospitalization for uncontrolled blood pressure.
*Medication use* Number of the medicines. Polypharmacy. Lack of adherence to treatment. Self-medication. Comprehension of the medication use.

**Table 2 ijerph-20-05233-t002:** Number of excerpts coded by theme.

Themes	n
Medication use	191
Comorbidity	97
Lifestyle habits	51
Health care utilization	41
**Total**	380

## Data Availability

For ethical and privacy reasons, the research data is not available.

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
