# Peer review of "Professionals’ and Patients’ Perspectives on Criteria for Referring Hypertensive Patients to Comprehensive Medication Management Services in Public Primary Health Care"

_ijerph, 2023, doi:10.3390/ijerph20075233_

Round 1
Reviewer 1 Report
Dear Authros,
Lifestyle habits of patients influence the development and control of hypertension. Patients need holistic care, i.e. taking into account their personal beliefs, values and culture, can help achieve therapeutic goals. The presented research can help develop tools and methods to provide holistic care and support to patients with hypertension. However, based on the following studies, conclusions can only be drawn for patients who are beneficiaries of the same health care system.
The references is quite old for this topic, it is worth updating using the latest publications.
Author Response
Dear reviewer,
We appreciate the opportunity to improve our article for possible publication. Your comments and suggestions were helpful in making the article more interesting for readers. We accept all suggestions made, as highlighted below.
We are at your disposal for any additional changes that may be necessary.
As suggested, the references have been updated and are highlighted in red.

Reviewer 2 Report
The authors want to create some kind of screening survey or instrument that will aid medical providers to make decisions about patients that should be part of a medication management regimen or program with a pharmacist.
I have no doubt the information collected is valuable and will serve as the initial point to identify the factors and patient conditions to help in the decision-making process of the medication management program. However, the authors need to work on presenting the results of the study in a better manner. Factors identified should be clearly presented so it is easy to letter recall what they are, and how the physicians can get information from patients to be able to identify those factors. A list of identified factors should be included at the end of the result section. The excerpts from the interviews and focus groups are valuable and valid to include in the paper but they need to be better connected to the paper's structure and the speech. I suggest the use of other metrics obtained from the Atlas.ti® software that might be better to refer to as how the important factors were identified. As it is presented now, the research and all the information collected lack importance since it looks like nothing new was discovered from the analysis of the data apart from what can be identified from the literature.
I made some comments in the manuscripts. (highlighted and commented in each highlighted section).

Author Response
Dear reviewer,
We appreciate the opportunity to improve our article. Your comments and suggestions were helpful in making the article more interesting for readers. We accept all suggestions made, as highlighted below.
We hope that the article now meets the journal's quality criteria. We are at your disposal for any additional changes that may be necessary.
The answers are described below:
Your comments were accepted. The alterations in the article are highlighted in red.
As suggested, the authors improve the connections between the text and speech. These are highlighted in red.
To help in the comprehension of the results was added a chart that summarizing the main factors founded in the study.
The factors identified in the study coincide with what is already described in the literature. However, this is based on the perspective of patients and professionals in primary health care, which will support the development of an instrument for selecting patients with hypertension. Therefore, interviews and focus groups were carried out to corroborate such data. As can be observed in this study: Sousa, M. C.V.B.; Fernandes, B.D.; Foppa, A.A.; Almeida, P.H.R.F.; Mendonça, S. de A.M.; Chemello, C. Tools to Prioritize Outpatients for Pharmaceutical Service: A Scoping Review. Res. Soc. Adm. Pharm. 2020, 16, 1645-1657. (reference 30 in the article).
In line 89, you questioned about this term - decompensated clinical condition. The selection instrument to be built will be directed to this profile of patients.
About your sample size questioning, in qualitative research, the concept of saturation is used, when there is no addition of new data with data collection in a certain number of individual or group interviews. In this type of research, quality is more important than quantity. The articles below demonstrate this:
- de Souza Minayo, M. C. Amostragem e saturação em pesquisa qualitativa: consensos e controvérsias. Revista pesquisa qualitativa. 2017, 5, 7, 1-12.
- Sebele-Mpofu, F. Y. The Sampling Conundrum in Qualitative Research: Can Saturation Help Alleviate the Controversy and Alleged Subjectivity in Sampling?. Int'l J. Soc. Sci. Stud. (2021, 9, 11.

Reviewer 3 Report
Dear Authors:
Developing qualitative research represents one of the great challenges as researchers, your work complies with elements of structure and form, however, I have some observations in this regard:
1.- Abstract: Add more information about the methods
2.- Methods: although described, I consider that the number of participants is not sufficient to generalize and conclude the behavior of a population.
3..- Conclusion: Add recommendations for future studies.
Author Response
Dear reviewer,
We appreciate the opportunity to improve our article. Your comments and suggestions were helpful in making the article more interesting for readers. We accept all suggestions made, as highlighted below.
We hope that the article now meets the journal's quality criteria. We are at your disposal for any additional changes that may be necessary.
The answers are described below:
The number of participants and the data collection period were added in the abstract. This are highlighted in red.
About your sample size questioning, in qualitative research, the concept of saturation is used, when there is no addition of new data with data collection in a certain number of individual or group interviews. In this type of research, quality is more important than quantity. The articles below demonstrate this:
- de Souza Minayo, M. C. Amostragem e saturação em pesquisa qualitativa: consensos e controvérsias. Revista pesquisa qualitativa. 2017, 5, 7, 1-12.
- Sebele-Mpofu, F. Y. The Sampling Conundrum in Qualitative Research: Can Saturation Help Alleviate the Controversy and Alleged Subjectivity in Sampling?. Int'l J. Soc. Sci. Stud. (2021, 9, 11.
The topic was added in conclusion as suggested. This are highlighted in red.

Round 2
Reviewer 2 Report
Dear authors,
I was expecting a much bigger transformation of the paper, with special attention in the result section. You used the Atlas.ti® software to identify the important factors, then why you are not using or presenting any of the metrics obtained from the software to refer to that process. About presenting the factors in a clearer manner, the table is good but I think it would be better to see the table at the beginning rather than at the end of the section. My idea with me is help the reader structure the information presented in the paper. You didn't present information or say something about how the physicians will be getting information from patients so they (physicians) will be able to identify those factors. And once again, as it is presented now, the research and all the information collected lack importance since it looks like nothing new was discovered from the analysis of the data apart from what can be identified from the literature. You said my statement is true and that was why you did focus groups (to validate the information?) but then what is your contribution? You need to work on highlighting the need for the paper, or the need for the instrument, it is only to validate what has been said before.
Author Response
Dear reviewer,
Thank you for contribution and suggestions in the article.
Your comments were accepted. The alterations in the article are highlighted in red. The metrics obtained from the software Atlas.ti® to refer to that process were included (Tables 1 and 2).
Regarding to your questions we may say: Besides, the factors identified in the study coincide with what is already described in the literature, the association of the two data collection tools (interviews and focus group) allowed knowing the perspectives of all those involved in the care process (caregivers and patients) to develop a patient selection instrument for the CMM that is more faithful to the demands of users. The instrument developed will contain questions that cover the themes that emerged in this study and will be applied by health professionals (physicians, pharmacists, nurses) to users of Health Centers in Primary Health Care. And, as we know, there is no validated instrument that can select patients for CMM in PHC (Sousa, M. C.V.B.; Fernandes, B.D.; Foppa, A.A.; Almeida, P.H.R.F.; Mendonça, S. de A.M.; Chemello, C. Tools to Prioritize Outpatients for Pharmaceutical Service: A Scoping Review. Res. Soc. Adm. Pharm. 2020, 16, 1645-1657).
We have included a paragraph with a justification for this in the introduction to make it clearer.
Reviewer 3 Report
Dear Authors:
I appreciate the modifications you made to the paper, and likewise the explanation of the sample size.
Author Response
Dear reviewer,
thank you.